# Teachers' Perception of Some Effects of the COVID-19 Lockdown: The Case Study of Ludovika University of Public Service

Gábor László [1] , Nikolett Deutsch [2,*] and László Berényi [1,*]

1   Department of Public Management and Information Technology, Faculty of Public Governance and International Studies, Ludovika University of Public Service, H-1083 Budapest, Hungary; laszlo.gabor@uni-nke.hu
2   Institute of Entrepreneurship and Innovation, Corvinus University of Budapest, H-1093 Budapest, Hungary
*   Correspondence: nikolett.deutsch@uni-corvinus.hu (N.D.); berenyi.laszlo@uni-nke.hu (L.B.)

**Abstract:** The COVID-19 lockdown has had serious consequences, including rethinking higher education. The study aims to enhance the knowledge base of online education and academic integrity through a case study of the Ludovika University of Public Service (LUPS), Budapest, Hungary. The research aimed to assess the teachers' experience with distance learning and examinations, including the change in workload, digital competencies, Moodle, Turnitin, and other software used during and after the lockdown. This paper summarizes the university-level policy changes induced during the lockdown, covering the introduction of emergency distance teaching and online examinations in academic integrity at the university. Two years after the first lockdown, the researchers made a survey (n = 145) about the continuation of the introduced solutions. The results show that a remarkable reordering started while the technical and technological backgrounds were available for the changes. The teachers could feel a significant increase in workload with distance education and have low trust in maintaining the standards of academic integrity. However, the research shows moderate and low levels of digital competencies in the majority of teachers, which clearly defines the most crucial task leading to success. Maintaining the monitoring system with objective indicators of the development and the opinions of the interested parties is essential for successful strategies in the field.

**Keywords:** digital transformation; academic integrity; emergency remote learning; online examinations; COVID-19

## 1. Introduction

The sudden lockdown in 2020 demanded a rapid digital transformation, among other things, in higher education. The university decision-makers and teachers closely observed the consequences of the forced change in their working and teaching culture to maintain their operation (Nurhas et al. 2022). Digital transformation, in this sense, can be interpreted as a process where digital technologies create disruptions triggering strategic responses from organizations that seek to alter their value creation paths while managing the structural changes and organizational barriers that affect the positive and negative outcomes of this process (Vial 2019).

Although digital education is perhaps the most commonly mentioned impact of the lockdown, it is to be noted that the topic has not emerged with the COVID-19 pandemic; the origins are much older (Cross 2004; Hubackova 2015). The measures related to the lockdown just accelerated the process, leading to fundamental changes in a matter of weeks. Moreover, exploring and understanding the impacts requires considering several other factors, including national and personal maturity, traditions, and work organizations. Schmidt and Tang (2020) give an overview of the technology in education, covering the specificities of e-learning, online learning, blended learning, and mobile learning. The main aspects are discussed by Williams (2002), divided into three categories:

1.  Pedagogy: the challenge is to select and use digital technology to enhance learning. Pedagogical and organizational challenges related to designing, delivering, and evaluating online and blended learning experiences.
2.  Participation: the challenge is achieving the appropriate involvement of students (learners) and teachers (educators) in the new learning environments.
3.  Access: the infrastructural, equipment, and acceptance challenges and the acceptance of the new technologies.

However, the goal of achieving a higher level of digitalization in education or any other field of our life is common; the practical implementation is complex. The national-level DESI index (European Commission 2022a, 2022b) values mirror local cultural differences, legal regulations, or financial opportunities that require targeted development actions. It was assumed that differences within countries are significant: in higher education, some common rules were given, but each institution was allowed to find the most suitable solution. Due to this, unique case studies can contribute to enhancing the knowledge base about online education.

The study is based on the experience and a survey at a Hungarian higher education institution. Considering the limitations of generalizing the results due to the local differences, the paper presents a pilot-level case study. The experience may be valuable for any higher education institution in developing its practices. The subject of the case study in this paper is the Ludovika University of Public Service (LUPS), Budapest, Hungary. An additional contribution of the study to the knowledge base is that the university offers special and unique programs in governmental studies, law enforcement, and military sciences.

A UNESCO report (Abdrasheva et al. 2022) comprehensively analyzes the impacts of COVID-19 on higher education institutions, concluding that they need to be better prepared for future crises to ensure support systems are in place for an uninterrupted learning environment. The authors selected the topics in this paper for investigation in line with the UNESCO report but with a limited scope by considering the access to data and the local experience. Those cover the digital competencies of teachers, access to ICT tools, and access to the Internet, covering one set of factors influencing the quality of online education. Another question is the approach to teaching and learning, including time management and academic integrity issues.

The remaining parts of the paper are organized as follows. Section 2 summarizes the background, including the national and university-level regulations and the responses to the lockdown. Section 3 introduces the research methodology. Section 4 shows the survey results, followed by the conclusions.

## 2. Background

### 2.1. Emergency Distance Education and Reordering

Although the rapid digital transformation was forced, this kind of change offered great opportunities to establish new strategies for enhancing the study programs (Mosquera et al. 2022). The cost–benefit of online solutions is apparent, but several critical factors must be considered. Of course, course development must be aligned with labor market needs (Boustani 2023; Belchior-Rocha et al. 2022). Mohamed Hashim et al. (2022) emphasize that quality evaluation is required to investigate how digital transformation as a propelling force could be used to build competitive advantages for universities. Appropriate strategies in the field need a thorough situation analysis. Implementing such studies is still not too late, three years after the lockdown, especially as other effects and a significant reordering have since become known.

Understanding and managing the challenges of digitalized education requires a broad-based analysis, even in the case of a focus on integrity issues. The technological background, infrastructure, personal competencies, or maturity of regulation impact the processes and attitudes. Williams' (2002) findings still seem to give a valid baseline for analyzing digitalization in higher education in the pandemic situation:

1.  A lack of resources and infrastructure for supporting digital technologies and online platforms.
2.  Access barriers and a digital divide among students and faculty due to socio-economic, geographic, or cultural factors.
3.  A lack of skills and digital literacy among students and faculty in using digital technologies effectively and responsibly.
4.  Resistance to change and innovation from some stakeholders who might prefer traditional or face-to-face modes of education.
5.  Ethical, legal, and quality issues related to data protection, privacy, security, plagiarism, assessment, and accreditation.

According to emergency distance education, Hodges et al. (2020) underlined that the curriculum was not re-designed for online use; in addition, the methodological preparedness of the teachers and the institutions was incomplete. In addition, the missing or limited changes in methodology were typical, which can be explained by the sudden change. It is, therefore, a response to an unexpected need, which places a heavy workload on each stakeholder (students, teachers, administrative, and IT staff). One more critical factor is the communication providing constant feedback and keeping the system functioning accordingly (Mohmmed et al. 2020).

Of course, the full participation of both students and teachers is desirable in order to enjoy the benefits of the new technology. Beyond the question of capabilities, an acknowledgment of the increased time and effort required in the field may be critical (Williams 2002), which is a highly sensitive issue in the forced environment caused by the COVID-19 pandemic. The reason for this is that without committed teachers, also playing the role of curriculum developers, the future improvement and enhancement of the applications will be put at risk. A piece of evidence is the fast reordering to face-to-face modes of education, and especially exams, as soon as it became possible.

### 2.2. Bringing Academic Integrity to the Fore

Regarding the distance forms of education compared to the traditional way, academic integrity failures have been among the main concerns for ages. Nuss (1984) noted that cheating and competition are well-known in higher education and valued above academic integrity. A broad range of methods have long been known to provide unethical advantages, including using cheat sheets, copying tests and essays from classmates, plagiarism and falsifying data (Comas et al. 2011), or using external contacts to communicate the answers secretly (Lancaster et al. 2019). The role of the Internet has also been appreciated in finding the correct answers (Comas-Forgas et al. 2021). The latter forms became particularly easy to implement during the lockdown. If a university wishes to maintain its high standards and the confidence of the labor market, the related processes, particularly the causes, must be managed.

Macfarlane et al. (2014) emphasized that much of the literature tends to focus on the negative framing of the topic and reported a perceived lack or absence of academic integrity. Academic integrity is a complex phenomenon; a comprehensive analysis of the terms goes beyond the scope of the study. The European Network for Academic Integrity (ENAI) definition of academic integrity is "compliance with ethical and professional principles, standards, practices and consistent system of values that serves as guidance for making decisions and taking actions in education, research and scholarship" (Tauginienė et al. 2018).

A former study (László 2020) confirmed that academic integrity was a critical issue even before the online solutions of the pandemic lockdown. The study was conceived to fill a gap in the information on academic integrity at the national level of Hungarian higher education, paying attention to the currently implemented anti-plagiarism practices and software. The results and correlations shown by the study indicate that there is much to be done in this field (László 2020). According to the lockdown in 2020, a formal assessment of the impact of the online examination policies and regulations on academic integrity was missing. The questions of the former study have remained relevant (László 2022).

McCabe et al. (2001) analyzed contextual and individual factors behind cheating to establish effective prevention. Although "cheating" is a plain and generally accepted term, the definition of cheating and unpermitted behaviors has been refined since that time. The forms of unpermitted behaviors are summarized as academic misconduct or academic dishonesty by universities, and these are published with similar content, including gaining or attempting to gain unfair benefits. The forms include plagiarism, contract cheating, collusion, impersonation, and misrepresentation. At the same time, misconduct goes beyond teaching and learning problems. Among others, such practices may also appear in journal publication practices, falsified grades or degrees, bribes to staff, teaching and research quality issues, or the falsification of faculty records (Stone and Starkey 2011). This paper focuses on student issues in the mirror of teachers' perceptions and deals with two types of misconduct. First, cheating is used as an umbrella expression for unpermitted individual or collaborative solutions of the students in tests, exams, and homework. Second, plagiarism is highlighted as a serious issue. It must be noted that the data collection was performed before the explosive spread of using Artificial Intelligence (AI)-supported software for cheating, but all this confirms the relevance of the problem.

### 2.3. National and University-Level Responses to the Lockdown

The Decree 40/2020. (III. 11.) of the Hungarian government on the declaration of a state of emergency as a consequence of the new coronavirus (SARS-CoV-2) pandemic outbreak laid the foundations for actions by higher education institutions as well. Government Decree 41/2020 (III.11.) on the measures to be taken during the state of danger declared that, for the prevention of the human epidemic endangering life and property and causing massive disease outbreaks, for the elimination of its consequences, and for the protection of the health and lives of Hungarian citizens, students were prohibited from entering higher education institutions.

It is common sense that the forced emergency responses boosted the digital transformation in higher education. However, it was not a complete transformation, just a transition to "emergency remote teaching" without common standards or central guidelines. Local responses to the challenges attempted immediate solutions with the available resources, including the IT background, human resources, and regulations. Some universities allowed the teachers to seek appropriate solutions, while others, including the Ludovika University of Public Service, strictly limited and controlled the available on-campus software applications. The changes in the approach at the university were reported by László (2022).

The university switched from attendance education to remote teaching within ten days during the first lockdown in 2020, between the 12th of March and the 22nd of March. That time was declared an educational break, and the staff used it to prepare for distance learning. Blended or synchronous online classes were not allowed in the spring semester; a full online education was ordered. In practice, that usually meant pre-recorded lectures and online submitted assignments via Moodle or email. The knowledge assessments (including mid-term and end-term exams) for all subjects were based on the evaluation of students' assignments (essays) prepared at home (Koltay 2020c). According to that, the Turnitin software (for similarity analysis) was available but not mandated and promoted; only a few teachers used it voluntarily (László 2022). It is a support tool for identifying text similarity that may constitute plagiarism (Foltýnek et al. 2020) and a relevant input for making the teacher's decision on the issue.

The next academic year (2020/2021) did show a quick reordering in education with face-to-face teaching and the exclusion of online exams. At the same time, the regulations have maintained special conditions and preparations for the transition to online teaching again. The new lockdown in November led to a changeover to online education (Koltay 2020a), including online exams and different submissions. The Study and Examination Regulations were adjusted to this situation two times, on the 21st of November and the 23rd of December (NKE 2020; Koltay 2020b). Beyond the online oral exams, Moodle offered a framework for written tests (with the built-in text editor), essay submissions (uploads),

and online tests (Quiz), (LUPS 2020). For the spring semester of the 2021/2022 academic year, teaching and assessment were performed online, with only minor refinements in the regulations (Koltay 2021; LUPS 2021; László 2022).

Actually, most of the regulations are outdated at the university. Online lectures and materials are used as supplementary learning sources, and distance exams are not allowed.

### 3. Research Goals and Methodology

*3.1. Research Goals and Limitations*

The study aims to contribute to enhancing the knowledge base of online education and academic integrity through the pilot case study of the Ludovika University of Public Service, Budapest, Hungary. The experience of an absentee-online education offers relevant information for the future since the change in workloads, new ways of achieving the satisfaction of teachers and students, and recent IT and methodological solutions have brought changes far beyond temporary use.

The research goal was to assess the experience of teachers with distance learning and exams, including the impacts on workload and student behavior. On the other hand, it provides the university with further directions for development in relevant areas based on practical experience and research results.

The research design used a systematic approach for a comprehensive overview, including teachers' competencies, situational factors, the utilization of software, and opinions along three research questions formulated:

- RQ 1. What is the preparedness of the teachers for digital education?
- RQ 2. What is the experience of the teachers with distance education and online solutions?
- RQ 3. What is the experience of the teachers with academic integrity issues?

RQ 2 covers the issues of changes in the workload, the experience with online education, and exams and supporting tools used during that period.

There are differences hypothesized in the experience with distance education within the university.

Although a thorough research design was applied in the study, some limitations must be mentioned. First, the case study of one university cannot represent higher education at a national level. The generalization of the results is not supported. Second, a self-managed online survey was used for data collection; a bias in the responses must be considered. Voluntary participation was allowed in answering a question, and as a result, a low response rate could be expected, and the representativeness of the sample was not ensured.

*3.2. Data Collection and Analysis Method*

A survey instrument was designed with an online, self-managed questionnaire for the teachers at the Ludovika University of Public Service, the content of which can be used directly by other universities.

The questions have endeavored to use scale evaluation on a five-point scale to make statistical analysis simple, but multiple-choice and open-ended questions were also used for grouping and collecting qualitative information where applicable. Investigations covered the teachers' opinions and experience during and after the pandemic as of the spring semester in 2022. According to the research questions, the survey formulated questions about the preparedness of the teachers, experiences with distance education, and the perception of academic integrity issues.

The preparedness of the teachers (RQ 1) was measured by the selected items of the self-evaluation of digital preparedness based on some competency areas of DigCompEdu (Redecker and Punie 2017):

- 1.3 Professional Engagement—Reflective practice (Dig1)
- 2.1 Digital Resources—Selecting digital resources (Dig2.1)
- 2.2 Digital Resources—Creating and modifying digital resources (Dig2.2)
- 3.2 Teaching and Learning—Guidance (Dig3)
- 4.1 Assessment—Assessment strategies (Dig4)

- 5.2 Empowering Learners—Differentiation and personalization (Dig5)
- 6.5 Facilitating Learners' Digital Competence—Digital problem-solving (Dig6)

The experience with distance education and exams (RQ 2.) included items for the general experience and the use of tools:

- the change in workload during and after the lockdown in the fields of education and administration,
- the experience with online education and exams,
- the tools and equipment used during and after the lockdown (with a highlight of the Moodle system).

The field of academic integrity was assessed based on the experience of the teachers, including questions about

- cheating during the exams,
- cheating and plagiarism in homework assignments,
- the experience with text-matching software (Turnitin).

The survey included the informal consent of the respondents.

Beyond the survey, the objective usage data of the university's learning management system (Moodle) was available. Its analysis is presented in a separate section.

The survey design allowed for statistical analysis. Since the goal was to explore the characteristic patterns, descriptives and first-generation explanatory (cluster analysis) and confirmatory (crosstabulation) techniques were applied (Hair et al. 2022). The statistical analysis was supported with IBM SPSS software, following the guidance of Pallant (2020). All statistical tests are presented at a 95% confidence level.

The cluster analysis was used to assess the digital competencies measured by the selected DigCompEdu questions. Since the purpose of the study was to explore the patterns through the group of respondents without prior grouping assumptions, the hierarchical clustering procedure was applied with the Ward method to minimize the increase in the total within-cluster sum of squared errors. Due to the result of the dimension reduction (principal component analysis with Varimax rotation) that offered only one factor with an eigenvalue over 1.0, the original questions and the correlations between the responses were significant in each case; the original items were used for the analysis.

The statistical significance of crosstabulation was tested by Pearson's Chi-square indicator. According to the hypothesized differences by faculty affiliation and checking the explanatory power of other grouping factors, a non-parametric variance analysis (Kruskal–Wallis H-test) was performed. It is to be noted that the study did not find significant grouping factors in the sample; accordingly, the related results are not presented.

*3.3. Sample Characteristics*

The Ludovika University of Public Service is a Hungarian state university with four faculties located in Budapest and Baja. The number of students in the spring semester of 2020 was 5479, and in the fall semester, it was 5908. During the same period, the number of teachers was 731 and 853, respectively.

The survey used anonymous and voluntary data collection. The university granted access to the email addresses of the target audience, but responses were not compulsory to avoid distortions in the answers. The invitation was sent to 1129 teachers (lecturers, educators, and trainers) of the university through the official NEPTUN Education System on the 19th of April 2022 (Table 1). Data collection was closed on the 30th of May with 145 responses.

Among the respondents, academic people with a researcher job position could have been more active in their responses. In total, 94.5% of the responses belong to teachers in different positions. The applicability of distance education is different based on the fact that the content is rather theoretical or practical. Of course, presenting theories, solving a mathematical example, or doing physical education can be implemented differently online. Table 2 summarizes the distribution of the respondents by focusing on their activity.

**Table 1.** Invitations were sent and surveys were filled by faculties.

| Faculty | Invitation Sent | Survey Filled | % within the Sample |
|---|---|---|---|
| Faculty of Public Governance and International Studies | 283 | 41 | 28.3 |
| Faculty of Military Science and Officer Training | 268 | 46 | 31.7 |
| Faculty of Law Enforcement | 257 | 31 | 21.4 |
| Faculty of Water Sciences | 61 | 27 | 18.6 |
| Not-specified staff members | 260 | | |
| Total | | 145 | 100.0 |

**Table 2.** Respondents by positions and teaching orientations.

| | Group | Frequency | Percent |
|---|---|---|---|
| | professor or associate professor | 55 | 37.9 |
| Position | lecturer, assistant lecturer | 82 | 56.6 |
| | research professor | 5 | 3.4 |
| | researcher | 3 | 2.1 |
| | theoretical | 63 | 43.4 |
| Orientation of teaching | both | 57 | 39.3 |
| | practical | 25 | 17.2 |

## 4. Results and Discussion

### 4.1. Learning Management System Used during the Pandemic

The increase in Moodle system use is remarkable after the lockdown. The primary learning management system at the Ludovika University of Public Service has been Moodle since the establishment of the university in 2012 (László and Szakos 2022). The system has also been used at the predecessor institutions. Moodle is often mentioned as the most popular open-source learning management system, supporting full-online or online blended education elements. Gamage et al. (2022) emphasize that Moodle is a popular choice for adaptive and collaborative learning and is increasingly used in online assessments. Moodle is continuously developed to address academic integrity, ethics, and security issues, enhance speed, and incorporate artificial intelligence (Gamage et al. 2022).

As appropriate, using Moodle at the Ludovika University of Public Service was a key element in responding to the lockdown challenges. A former study (László and Szakos 2022) described the use of the system by uploaded learning assets. The tendencies (see Figure 1) underline the increase in use and also show that there was a multistep enhancement, which is consistent with the adoption categories (early adopters, early and late majority) in Rogers' diffusion theory (Rogers 1995).

A new version of the Moodle learning management system has been introduced at the university to improve the learning processes in the 2021/2022 academic year. The new Moodle has expanded functionality and renewed design using the educational experience of the lockdown period.

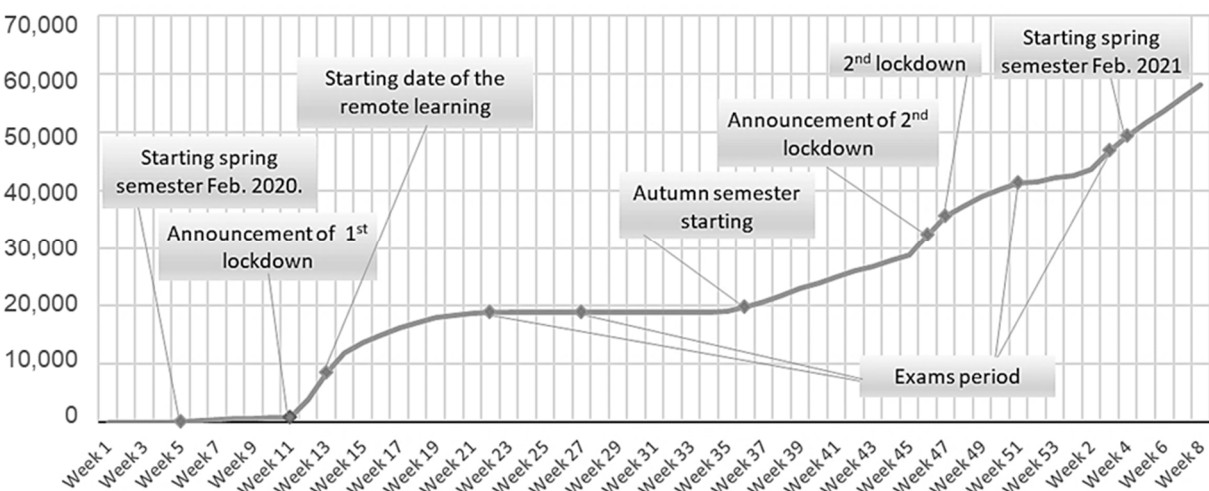

**Figure 1.** Number of uploaded resources (files, links) between the semesters 2019/20/2 and 2020/21/2 (based on László and Szakos 2022).

*4.2. Teachers' Competencies*

The availability of hardware and software for online and distance education is useless without prepared and committed users on both sides of education. The Digital Competence Framework for Educators of the European Commission (DigCompEdu) allows for finding gaps and improvement opportunities in related personal competencies (Redecker and Punie 2017). The survey used the original wording of the selected DigCompEdu items.

The analysis used the ranking of the statement on a six-point scale. Each question listed the related descriptions of the DigCompEdu items for the six maturity levels, and the respondents had to choose what best describes them. The values in Table 3 present the mean values of the rankings; the higher numbers represent a higher level of competencies in the field. The results show moderate mean values of the teachers' competencies between the integrator (B1) and expert (B2) levels.

**Table 3.** Mean values of DigCompEdu results by the clusters.

| Ward Method | | Dig1 | Dig2.1 | Dig2.2 | Dig3 | Dig4 | Dig5 | Dig6 |
|---|---|---|---|---|---|---|---|---|
| Total sample | Mean | 3.31 | 3.70 | 2.33 | 2.88 | 2.50 | 2.88 | 2.90 |
| | N | 145 | 145 | 145 | 145 | 145 | 145 | 145 |
| Cluster 1 | Mean | 5.00 | 4.85 | 4.85 | 4.85 | 4.77 | 4.85 | 5.85 |
| | N | 13 | 13 | 13 | 13 | 13 | 13 | 13 |
| Cluster 2 | Mean | 2.49 | 2.55 | 1.58 | 2.17 | 1.89 | 2.04 | 1.81 |
| | N | 53 | 53 | 53 | 53 | 53 | 53 | 53 |
| Cluster 3 | Mean | 3.58 | 4.28 | 2.42 | 3.03 | 2.53 | 3.13 | 3.14 |
| | N | 79 | 79 | 79 | 79 | 79 | 79 | 79 |

Since the statistical analysis did not show significant differences by faculties or other grouping factors, a clustering process was applied to explore patterns of the teachers' preparedness. The procedure resulted in three clusters. However, this result has a limited interpretation; it allows for an exploration of the majority and minority patterns of the approaches with a highlight of the key competencies. The results described by the mean values in the clusters are shown in Table 3 and Figure 2.

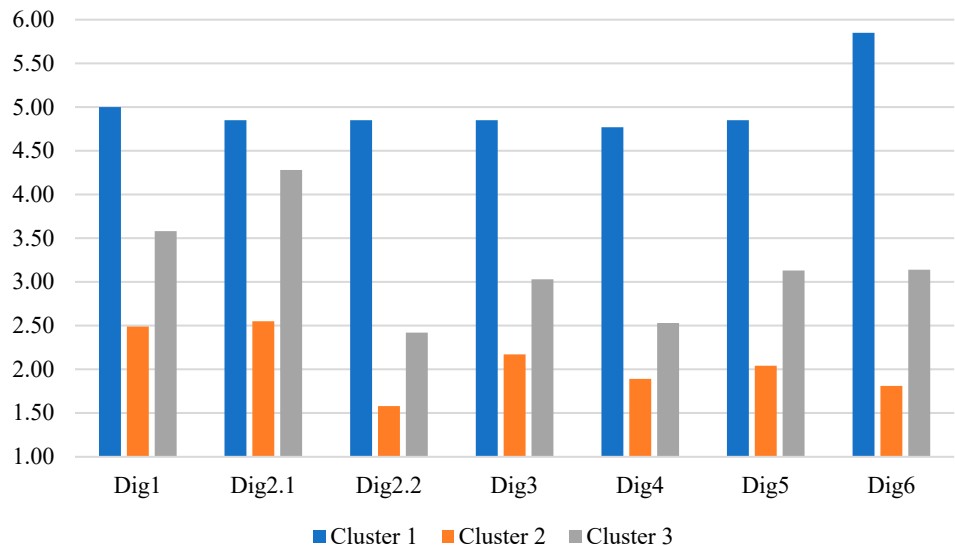

**Figure 2.** DigCompEdu results by clusters (ranking on a six-point scale).

The three clusters represent high, low, and moderate levels. Cluster 1 includes teachers with a high average level of competencies in each field, but only 8.9% of the respondents belong to this cluster. Cluster 3 is for the moderate level of competencies, including 54.5% of the respondents. Teachers with low competencies are in Cluster 2; they are 36.5% of the respondents. There are serious challenges to be noted in the mirror of the expectations and the determinative role of the teachers on the level of digital education.

*4.3. Workload Change*

The changing workload can describe situational issues due to regulations related to the lockdown. The related question was formulated as "How has the shift to online education changed the proportion of the following activities?", and "less", "no change", or "increased responses" could be selected. The lockdown and the distance education led to an increased workload in education and administration based on the teachers' responses, while the student consultation and the research task were increased by fewer of them (Table 4).

**Table 4.** Evaluation of the change in workload by the COVID-19 pandemic lockdown (% of the respondents).

| Workload | Education | Consultation | Research | Administration |
|---|---|---|---|---|
| Less | 4.8 | 22.8 | 26.2 | 4.1 |
| No change | 35.9 | 46.2 | 49.7 | 32.4 |
| Increased | 59.3 | 31.0 | 24.1 | 63.4 |

The crosstabulation analysis of the changes did not show significant differences by the faculty or the activity of the respondent, except for the administration (Pearson Chi-Square = 21.980, $d_f$ = 6, sig. = 0.001). In the cases of military education and the water science faculty (this one is located in another city, Baja), more teachers feel the increase in workload than in the cases of governmental and law enforcement teachers (Table 5).

There were 40 substantive responses to the optional question, marked as an open-ended question. It was pointed out that it took much more time to learn the online teaching methodology, to learn the teaching software, to prepare the online teaching materials (the materials had to be prepared in a different format to those previously taught), and to prepare the online examinations. Communicating with students consumed a significant amount of time due to the numerous individual and group messages that necessitated repeatedly

reiterating and clarifying the same information. Reducing the time spent traveling was also highlighted by many respondents.

**Table 5.** Change in administration workload by faculties (% of the respondents).

| Workload | Faculty of Public Governance and International Studies | Faculty of Military Science and Officer Training | Faculty of Law Enforcement | Faculty of Water Sciences |
|---|---|---|---|---|
| Less | 9.8 | 2.2 | 3.2 | 0.0 |
| No change | 43.9 | 19.6 | 51.6 | 14.8 |
| Increased | 46.3 | 78.3 | 45.2 | 85.2 |

*4.4. Software Use during and after the Lockdown Period*

Although a broad range of software was available, especially at the beginning of the lockdown, a free selection was not allowed (in fact, the use of certain practices was prohibited) due to the missing decisions on standardization, and the utilization of them was low; additionally, the rector's measure did not allow the opportunity for synchronous online classes in the spring semester of 2019/2020, and the software was reduced to Moodle, email, the NEPTUN Education System, and online webinars.

The survey included some popular solutions for symmetric or asymmetric communication channels with the students and each other. The survey included a question of whether the respondent used the service and another one concerned satisfaction. The user satisfaction was measured with a five-point scale. Figure 3 shows the ratio of the users and the ratio of the satisfied users combined. A satisfied user is defined as a respondent who marked four (satisfied) or five (very satisfied) values to the given question. The ratio of satisfied users is measured within the users of the given service.

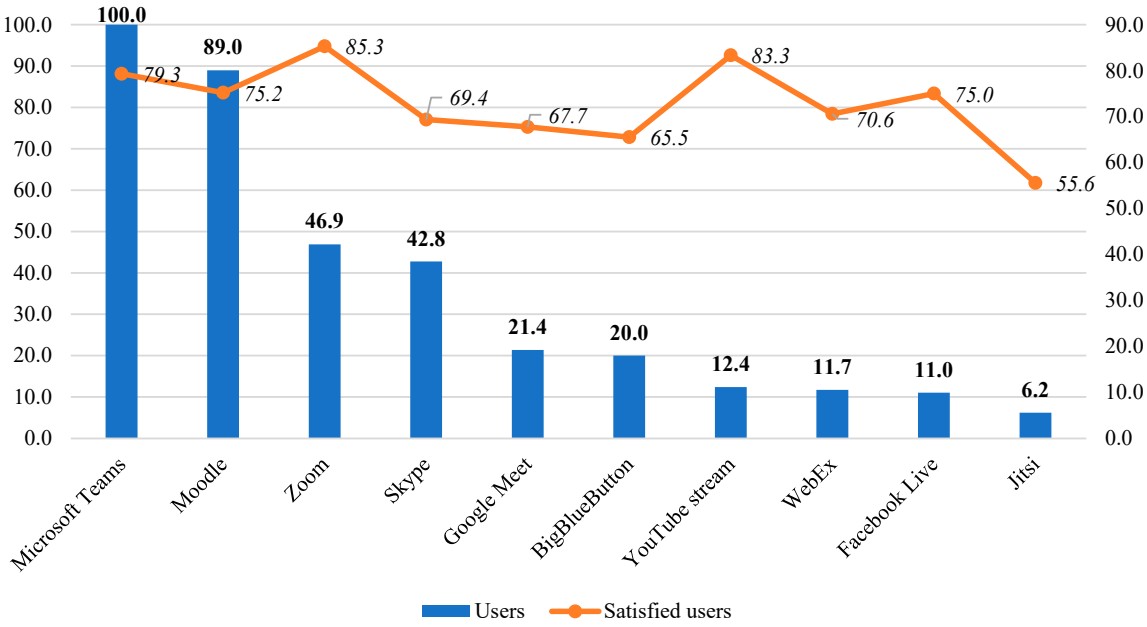

**Figure 3.** The ratio of users and, within that, the satisfied users of some services (% of respondents in the case of users, and % of the users in the case of satisfied users).

The "winner" among the software and services was Microsoft Teams. It must be noted that the university selected Office/Microsoft 365 as an official way of administration and communication in the fall semester of 2020/2021. After a short period of transition in the first months of the lockdown, Microsoft Teams was designated as the mandatory tool.

Zoom and Skype are also popular solutions, while others, e.g., Google Meet, WebEx, or even YouTube Stream, lag behind. The list included Moodle, which serves broader purposes than other items but belongs to education and is usually used in combination with others. Although Moodle is the official learning management system of the university, 11% of the respondents have not used it. According to satisfaction, more frequent use does not mean a better judgment of the services. Zoom or YouTube streaming led to higher satisfaction than Microsoft Teams among the teachers.

The use of Moodle shows a remarkable change comparing the before-lockdown and the later situations: teachers generally use the system. The related question asked to mark whether the respondent used Moodle "frequently", "rarely", or "not" before, during, and after the lockdown. Before the lockdown, 31 of 137 teachers were frequent users of Moodle; then, during the pandemic, their number increased to 101. After the lockdown, the number of frequent users fell back to 74 teachers. The statistical analysis shows faculty differences; the Faculty of Water Science is not a common user. The cross-tabulation is significant for the before-pandemic situation (Pearson Chi-Square = 18.188, $d_f$ = 6, sig. = 0.006), during the pandemic (Pearson Chi-Square = 61.533, $d_f$ = 6, sig. = 0.000), and also the after-pandemic situation (Pearson Chi-Square = 36.725, $d_f$ = 6, sig. = 0.006). By the position and the theoretical or practical orientation of the teachers, the analysis did not show different patterns.

Before the lockdown, the use of Moodle was voluntary, and it was mainly used as a supplementary framework for attendance education. The Faculty of Public Governance and International Studies played a leading role in the application. According to the Faculty of Law Enforcement, some teachers intended to use Moodle, but an official license would have been required.

Text-matching software (Turnitin is used at the university) is the most essential IT tool for supporting the academic integrity of students' work. The Ludovika University of Public Service has access to the web-based service, and the Moodle integration is allowed. The latter option offers an automatic review of student papers and receives the results in Moodle without additional steps in the Turnitin system. The number of Turnitin users among teachers is relatively low (Table 6). The analysis shows significant differences by faculties (Pearson Chi-Square = 10.522, $d_f$ = 3, sig. = 0.015) and by position (Pearson Chi-Square = 4.259, $d_f$ = 1, sig. = 0.039). The use of Turnitin integrated with Moodle is not spread. Based on the grouping factors, the faculty level differences are significant (Pearson Chi-Square = 11.034, $d_f$ = 3, sig. = 0.012).

**Table 6.** Use of Turnitin (number of respondents).

| | | Turnitin Use | | Turnitin Use in Moodle | |
|---|---|---|---|---|---|
| | | **Yes** | **No** | **Yes** | **No** |
| Total sample | | 39 | 98 | 25 | 112 |
| Faculty | Faculty of Public Governance and International Studies | 15 | 24 | 9 | 30 |
| | Faculty of Military Science and Officer Training | 17 | 27 | 13 | 31 |
| | Faculty of Law Enforcement | 4 | 27 | 3 | 28 |
| | Faculty of Water Sciences | 3 | 20 | 0 | 23 |
| Position | professor or associate professor | 21 | 34 | 12 | 43 |
| | lecturer, assistant lecturer | 18 | 64 | 13 | 69 |
| Orientation | theoretical | 18 | 39 | 11 | 46 |
| | both | 18 | 37 | 13 | 42 |
| | practical | 3 | 22 | 1 | 24 |

Although a Turnitin subscription has been available at the university for years, it only had limited access regarding the number of uploaded documents. The frame was reserved for checking Ph.D. Theses and book publishing. Most teachers did not have any information about the opportunity. Similarly, Turnitin–Moodle integration has also been self-taught by the teachers. The subscription has been extended, and broader communication has also been developed, which led to an increase in the application, but this is not yet the norm today. Further actions are needed, especially in communicating the opportunities and the training for use.

### 4.5. Detection of Plagiarism and Cheating

Cheating, plagiarism, and the efforts to prevent them have always been present in academic life; only the tools are refined with technological development. Both institutional policies and teachers' approaches are essential for detecting inadmissible instruments. The lockdown and the distance education raised the question to a new level. The situation may have been accompanied by increased fraud since detection and proof were more difficult online than in the classroom, and getting punished could be avoided more easily than before.

The mean value of the responses about the estimated ratio of cheating students during distance education was 43.7% (the standard deviation is 31.9). The ratio of plagiarism is estimated to be lower; the mean value is 31.9% (the standard deviation is 29.1). At the same time, 9.7% of them noted that cheating was found in the respondents' practice, and 32.4% found plagiarism. The responses justify our assumption about the increase in cheating, while the majority of the respondents said that there is no change in plagiarism (Figure 4). The statistical analysis shows a uniform approach of the teachers; there are no significant differences found in the distributions.

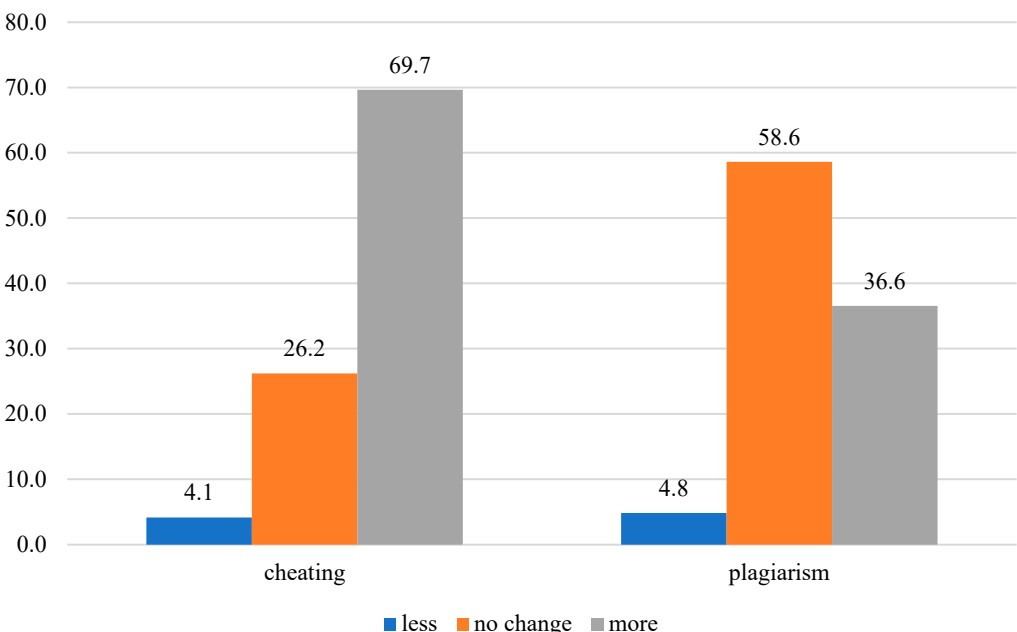

**Figure 4.** Change in student cheating and plagiarism during distance education in teachers' view (% of respondents).

The crosstabulation (Table 7) between the questions about cheating and plagiarism and the fact that the teacher has found any issues shows significant differences (for cheating: Pearson Chi-Square = 26.940, $d_f$ = 4, sig. < 0.001, and for plagiarism: Pearson Chi-Square = 13.253, $d_f$ = 2, sig. = 0.001).

**Table 7.** Crosstabulation results about cheating and plagiarism (distributions in % by yes and no answers).

| | | Did You Find Cheating Issues? | | Did You Find Plagiarism Issues? | |
|---|---|---|---|---|---|
| | | Yes | No | Yes | No |
| Assumed change in cheating/plagiarism issues | less | 7.1 | 1.5 | 4.3 | 5.1 |
| | no change | 7.1 | 10.8 | 38.3 | 68.4 |
| | increased | 85.7 | 87.7 | 57.4 | 26.5 |

*4.6. Responses to the Research Questions*

According to the research question about the preparedness of the teachers for digital education (RQ 1), the digital competencies related to the use of ICT tools in education and other educational activities are essential. The survey results show a lack of digital competencies in general. The development of digital pedagogical competencies has become even more important in a dynamically changing context, like the disruptive effects of artificial intelligence across the whole higher education spectrum. Both the COVID lockdown period and the present research have shown that digitization cannot be equated with transferring existing processes to the digital environment but requires rethinking and adapting processes to the online space and applying new methodologies. The unfavorable ranking position of the county by the DESI index suggests that the transformation requires special processes, and international examples may not be enough. The results of the DigCompEdu assessment confirm the assumption at the individual level. A remarkable proportion of the respondents show low competencies in digital issues.

The investigations associated with the research question about the experience with distance education and the related tools (RQ 2) allow us to highlight the increased workload of the teachers. In addition to the time spent on preparation and training, the time spent on administration has increased considerably. Regarding how the university handled distance learning and examinations, most of the teachers, based on their former personal experiences and personal preferences, considered it to be good.

In the context of the survey presented in this paper, according to the academic integrity issues (RQ 3), the text responses to the optional open-ended questions could provide valuable information for completing the statistical analysis. Two of the text responses highlight the relevance of this approach: "Students immediately could understand the weaknesses and backdoors of online education"; "I assumed that the students would use any tools or support, so I focused on the skills instead of the lexical knowledge". The use of proctoring tools was requested by some teachers, but there was also a legitimate concern for the student's privacy. The data show that integrity issues were followed more closely than before, and this emphasis on the topic has remained after the lockdown.

Among the problems related to distance education, the responses pointed to the importance of clear and timely communication, in addition to technical barriers and insufficient digital competencies. Teachers agreed that online teaching and exams had facilitated the use of unauthorized solutions by students during their tasks. In many cases, the teachers did not even know the technologies the students used (e.g., shared Discord groups).

## 5. Conclusions

The COVID-19 pandemic has brought unknown challenges for everyone in everyday life and higher education. The situation has also highlighted and, in many cases, magnified existing social inequalities, e.g., digital competencies or equal opportunities in terms of access. It has drawn attention to structural challenges, which have been addressed and resolved in different ways throughout closures. By the time divergent and sometimes inconsistent regulations of the institutions and faculties were established, the closures were unlocked. The need for training the trainers on new digital technologies has increased

(Papademetriou et al. 2022), as well as the formation of attitudes in the field of academic integrity. Strict regulation and a well-established value system grant the quality of education and learning, which directs more attention to the necessity of quality assurance.

In many aspects, the decreased use of digitalization after the lockdown was observed in both business and higher education. That could mean that some institutions in higher education might reduce or abandon their use of digital technologies and online platforms for teaching, learning, research, and administration. Dolenc et al. (2022) pointed out that teachers will return to traditional teaching when classrooms reopen. A holistic understanding of the causes of technological changes in the learning process is even more urgent (Zitha et al. 2023).

Referring back to the conclusion of the UNESCO report (Abdrasheva et al. 2022) about how higher education reflected the COVID-19 pandemic and the lockdown, it emphasizes the rapid return to the traditional teaching and learning methods, hindering a fundamental transformation in the core face-to-face endeavor understanding of higher education. The pilot case study presented in this paper investigated the changes and impacts at the Ludovika University of Public Service.

The responses confirmed the tendency that many teachers have returned to traditional teaching and face-to-face technologies. At the same time, an increasing number of them would like to incorporate positive pieces of evidence and change their teaching methods. The University of Public Service is an excellent example of an institution where these opportunities are enabled while strict regulation and a well-established value system grant the quality of education and learning. The low and moderate level of digital competencies among the majority of the teachers clearly indicates the most urgent fields for further improvements. The effective utilization of the technical background and the systems is not possible without skilled and committed human resources. A warning signal is the national performance by the Digital Economy and Society Index (DESI) index. Among the EU member states, Hungary ranked 23rd out of 27 EU Member States in the Digital Economy and Society Index country profile (European Commission 2021). That position was 22nd in 2022 (European Commission 2022a).

The trust in the new solutions is hindered in the sense of ethical behaviors. The teachers' perceptions suggest that online and distance ways of education amplified the use of unauthorized or fraudulent resources. Fraud detection was not impossible, but it was almost impossible to prove it clearly, especially later. Most of all, the teachers have sought to exclude cheating or at least make it more difficult to implement. It must be noted that the university policies limited the tools and opportunities for the teacher, and the software background did not include, e.g., proctoring that provides monitoring of the student's computer during an exam.

A methodological implication is maintaining the surveys about the teachers' perceptions in parallel with expanding the measurement through the objective indicators. Monitoring the tendencies may support effective strategy building.

A broader implication of the study is that just providing the technical background is not enough. The scattered picture of the practices even within one university and the mixed level of competencies suggest the need for the education of educators. Enhancing the investigations into other institutions may give a comprehensive picture of the preparedness and the gaps in the development.

**Author Contributions:** Author Contributions: Conceptualization, G.L.; methodology, L.B.; validation, N.D.; formal analysis, G.L. and L.B.; data curation, G.L.; writing—original draft preparation, G.L. and L.B.; writing—review and editing, L.B. and N.D.; visualization, L.B. All authors have read and agreed to the published version of the manuscript.

**Funding:** This research received no external funding.

**Institutional Review Board Statement:** Institutional Review Board does not exist at the University. Official authorization from the Dean's Office is available.

**Informed Consent Statement:** Informed consent was obtained from all subjects involved in the study.

**Data Availability Statement:** Research data are available to the researchers, as requested.

**Conflicts of Interest:** The authors declare no conflicts of interest.

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
