# Peer review of "Teachers’ Perception of Some Effects of the COVID-19 Lockdown: The Case Study of Ludovika University of Public Service"

_socsci, doi:10.3390/socsci13020111_

Round 1
Reviewer 1 Report
Comments and Suggestions for Authors
The article proposes an interesting topic but is very tiring to read. There are endless repetitions and redundancies that confuse the reader. It is not clear what is being measured and why.
In the end the work presents descriptive statistics that are devoid of scientific hypotheses on the buildings. I suggest the authors review the approach, following the scientific format and immediately clarify the objectives. As presented, does the work consist of clichés and does not report on scientific constructs? For example, we talk about academic integrity but this is not compared to previous measurement systems. The authors also propose correct reflections but without measurement systems such as the preparation of students as the final output of online and offline delivery.
​
Author Response
Dear Reviewer #1,
thank you very much for the appreciation of the topic and our efforts.
We apologize for the weaknesses of the manuscript. Although this is no excuse, originally, we planned a broader analysis of the topic, but due to some data quality issues and not exceeding the usual length of the papers in the field, some issues were left out. Meanwhile, the introduction and research design parts still involved references to these.
The manuscript is restructured and refined. On the proposal of another reviewer, the title is changed to give a better tone to the purpose and scope of the analysis.
We hope that you find the new version appropriate for further processing. Thank you very much for your thorough work and helpful advice.
The authors
Reviewer 2 Report
Comments and Suggestions for Authors
The case study is important and the research is significant. There is a need for clarity and coherence. Here there are some suggestions that could help:
1) the title could be more related coherently to the central topic of the article. After the introduction, which is a title that is addressed to the structure of the paper a title concerning the content is used. Could it be a part of the introduction? Please consider it (129 lines)
2} 244-255 lines refer to the aim of the research and 251-252 concerns methodology. and then there are the research questions. I suggest to define clearly the goal and the methodology issues
3) 271-272 refer to the content of the questionnaire.
4. 278-290 lines. The numbers have nothing to do with the research questions. So they need to be numbered differently.
5) Question 12,13,14,15 which the main subject have?
6) 227-230 has to be in the sample description 3.3.
7) in the method please present the statistical analysis
8) organize the research questions according to the results and the main issues of the questionnaire
9) the integrity is not the only issue of your research please reconsider the title
10) Use the theory background in interpreting your results
Author Response
Dear Reviewer #2,
thank you very much for your thorough work and the detailed notes on the critical elements.
Your suggestions highlighted the inconsistencies of the sections for us. We reorganized the structure of the manuscript following your instructions. That includes the goals and a new title, as well. Although this is no excuse, originally, we planned a broader analysis of the topic, but due to some data quality issues and not exceeding the usual length of the papers in the field, some issues were left out. Meanwhile, the introduction and research design parts still involved references to these.
According to the numberings, a technical problem may be in the background; after saving the file, it was always renumbered. We removed the automatic option for ordering. Hopefully, this can solve the issue.
We hope that you find the new version appropriate for further processing.
The authors
Round 2
Reviewer 1 Report
Comments and Suggestions for Authors he authors have significantly strengthened the work which can now be publishedAuthor Response
Dear Reviewer #1,
thank you very much for your valuable comments in the first review round.
We were happy to see that you could agree with the changes in the paper and that you accepted it
for publication.
Yours Sincerely,
The authors
Reviewer 2 Report
Comments and Suggestions for Authors
This is a better version of course. These are my suggestions for improvement
The font size in Tables 1.2.3. are larger than in the text.
In Figure 3 the font size is not appropriate to ensure the clarity of all elements of the figure
236 line 3.2. Data collection and analysis method
Comments on the Quality of English Language532-535 line _Very long sentence
538 line_Passive voice as it is more formal.
311 line_ Not clear language
311-314 line_ very long sentence
Generally, you could use more formal language
Author Response
Dear Reviewer #2,
thank you very much for your valuable comments in the review rounds. Thank you for your
appreciation of the improvements in the manuscript.
According to the suggestions in your review, we note that
- Table font size is checked and corrected in line with the template file.
- Figure 3 is adjusted.
- Subsection title (3.2) has been changed as suggested.
According to the grammar comments, the text was revised as suggested in your comments.
Yours Sincerely,
The authors